# An Overview of the Phytochemical Composition of Different Organs of *Prunus spinosa* L., Their Health Benefits and Application in Food Industry

Mariana Florica Bei [1,†], Alexandru Ioan Apahidean [2], Ruben Budău [1,†], Cristina Adriana Rosan [1], Raluca Popovici [1], Adriana Ramona Memete [1,*], Daniela Domocoș [3] and Simona Ioana Vicas [1,*]

1   Faculty of Environmental Protection, University of Oradea, 410048 Oradea, Romania; mbei@uoradea.ro (M.F.B.); rbudau@uoradea.ro (R.B.); crosan@uoradea.ro (C.A.R.); rpopovici@uoradea.ro (R.P.)
2   Faculty of Horticulture and Business in Rural Development, University of Agricultural Sciences and Veterinary Medicine, 400372 Cluj-Napoca, Romania; alexandru.apahidean@usamvcluj.ro
3   Faculty of Medicine and Pharmacy, University of Oradea, 410087 Oradea, Romania; ddomocos@uoradea.ro
*   Correspondence: adrianamemete@yahoo.com (A.R.M.); svicas@uoradea.ro (S.I.V.)
†   These authors contributed equally to this work.

**Abstract:** The purpose of this study is to analyze prospective approaches that emphasize the beneficial biofunctional and metabolic effects of different anatomic parts of the *Prunus spinosa* L. on maintaining human health and improving some pathophysiological processes. Our research is based on recent data from literature on the biology and ecology of *P. spinosa* L., focusing on its bioactive compounds. Based on such bifunctional parameters, we aim to conceptualize path-breaking approaches that have as a goal the incorporation of *P. spinosa* L. fruits into certain food products to make use of their potential metabolic benefits in cardiovascular pathologies and other disorders that are prevalent at present and respond to nutritional intake of antioxidants. It is well known that dietary interventions allow the search for mechanisms leading to the development of effective nutritional therapies. This review has mainly focused on the identification of bioactive and functional nutrients of *P. spinosa* L. that can be incorporated into diverse food formulations. This is because nutrition plays a pivotal role in the development, validation, and recommendation of the nutritional composition of food, with demonstrated impacts on metabolic processes in specific diet-related pathologies.

**Keywords:** *Prunus spinosa* L.; phenol compounds; oxidative stress; health benefits; food application

## 1. Introduction

Studies on the biologically active substances that are found in food are of great relevance in the innovation and certification of functional food products. Thus, at the global level, a great emphasis is placed on the search for plant-based foods as alternative options to prevent chronic inflammation through dietary interventions [1]. Food manufacturers aim to develop new food products that are attractive to a wide range of potential consumers while also trying to become competitive in the market. Currently, these actions can be divided into two directions, namely a return to natural and traditional products that are minimally processed and the production of functional foods, often using unconventional materials or additives. These directions have been established based on recent studies that prioritize the global effort to find technological methods for food processing that minimize the loss or degradation of biologically active phytocompounds. The fruits of *P. spinosa* L. contain high levels of phenolic compounds, which have strong anti-oxidant properties [2–5]. These compounds have potential applications in the food and phytopharmaceutical sectors [1,4,6–9]. The blackthorn can be utilized as an ingredient in various food products such as yogurt [10], ice cream [11], jam [12], wholemeal biscuits with dried fruit, gin and tonic drinks (the latter

being the focus of our research team). Incorporating blackthorn into these food items would enhance their nutritional value, therefore fortifying them and improving their overall quality. The recent study conducted by Özkan (2023) [13] demonstrated that dried *P. spinosa* L. pestles exhibit high bioaccessibility of polyphenols during gastrointestinal digestion. This finding suggests that *P. spinosa* L. could be a promising option for producing beverages. The utilization of *P. spinosa* L. extracts as novel anthocyanin-based food dyes in confectionery items, such as topping on donuts and in "beijinho", a Brazilian biscuit product, has resulted in significant changes in nutritional content [14]. Additionally, there has been a significant improvement in anti-oxidant and antimicrobial properties, leading to the conclusion that *P. spinosa* L. extract has great promise as a natural food colorant for patisserie-confectionery products [14]. Recent studies have shown that eating food rich in phenolic compounds can prevent degenerative decline, cardiovascular disease (CVD), and cancer, mainly due to the anti-oxidant potential of these natural substances [4,15–18].

Modern phytotherapeutics emphasizes the benefits of consuming parts or products based on the *Prunus spinosa* L. shrub as it is considered a plant with functional nutritional and therapeutic properties, remarkable in various pathologies with increasing incidence. Studies to date have shown that the polyphenols, which are present in significant amounts in the fruits of *P. spinosa* L., are biofunctional components, including anthocyanins, phenolic acids, flavonoids, and coumarin derivatives [4,6,19,20].

Unexpectedly, several widely used antioxidants have been shown to exhibit pro-oxidant properties. Three factors can affect the function of an anti-oxidant, causing it to become a pro-oxidant. These factors are the concentration of the anti-oxidant in matrix environments, the presence of metal ions, oxygen, and its redox potential [21]. Natural antioxidants can eliminate free radical intermediates and regulate enzymatic activity by inhibiting both the early and late stages of carcinogenesis. Condello & Meschini's recent study confirms the impact of flavonoids derived from *P. spinosa* L., Trigno ecotype (Trigno M), on oxidative stress. The study examines how different concentrations of blackthorn flavonoids can either act as pro-oxidants or antioxidants in the human colorectal cell line, HCT116, during in vitro treatment [22].

The importance associated with food science and nutritional science is made obvious by the transformation of traditional remedies into nutraceutical, dietary supplement preparations based on the properties of phytochemicals as regulators of cellular signals.

Both in the European tradition and in other countries around the world, the fruits of *P. spinosa* L. have been considered a traditional phytotherapeutic remedy that could be used for the treatment of inflammatory disorders of the gastrointestinal and urinary tracts or of the respiratory system. Used locally, they can reduce the inflammation of the oral and pharyngeal mucosa. The fruit, the flowers, and the leaves of *P. spinosa* L. are considered useful phytotherapeutic products, as they can be included in the treatment of diarrhea or metabolic diseases such as diabetes, obesity, and associated cardiovascular pathologies [8,23,24].

Since oxidative stress is the main risk factor in several chronic and degenerative pathologies [7], researchers have put considerable effort into identifying and using natural antioxidants from both plants and edible berries so as to either prevent the onset or counteract the progression of the aforementioned diseases [2,7,25].

The results of the study on the anti-inflammatory and anti-oxidant effects of extracts from the fruits of *Prunus spinosa* L. on human immune cells in relation to the phytochemical profile, analyzed *in vivo*, support the traditional use of blackthorn fruits in inflammatory conditions and recommend these extracts as promising for obtaining functional food products [24].

As it is based on an overview of recent literature, the purpose of the present investigation is to assess the efficacy of blackthorn as a whole plant, highlighting the phytochemical profiles of each botanical part of this deciduous shrub. Furthermore, starting from the literature analysis, functional attributes have been attributed to each anatomical part of the

blackthorn shrub, with the aim of discovering nutraceutical uses in medicine and the food industry. A systematic review of the existing literature was conducted.

## 2. Research Methodology

PRISMA Flowchart 2020 was used to select data on the nutritional and phytochemical composition of *P. spinosa* L., as well as its biological activity, based on the proposal of Page et al., 2021. Figure 1 shows the stages and selection criteria, followed by the number of studies used in our review. The most recent research on *P. spinosa* L. has been obtained from PubMed, Scopus, Science Direct, Elsevier, Google Scholar, and Google patents. Overall, this review is based on up-to-date data that has been published within the last 5 years. The following keywords were included in the search: "blackthorn" "bioactive compounds *P. spinosa*", "phytochemicals *P. spinosa*", "anti-oxidant capacity/activity *P. spinosa*", "antimicrobial *P. spinosa*", and "*anticancer P. spinosa*". The data presented in the tables were obtained from recent research publications that include both in vivo and in vitro studies. *P. spinosa* L. studies published in languages other than English and Romanian were omitted. In addition, we eliminated excessively outdated research papers and those that did not generate interest, as well as those that diverged from the content on which we focused. A comprehensive selection process was conducted, resulting in the inclusion of a total of 162 studies in this review (Figure 1).

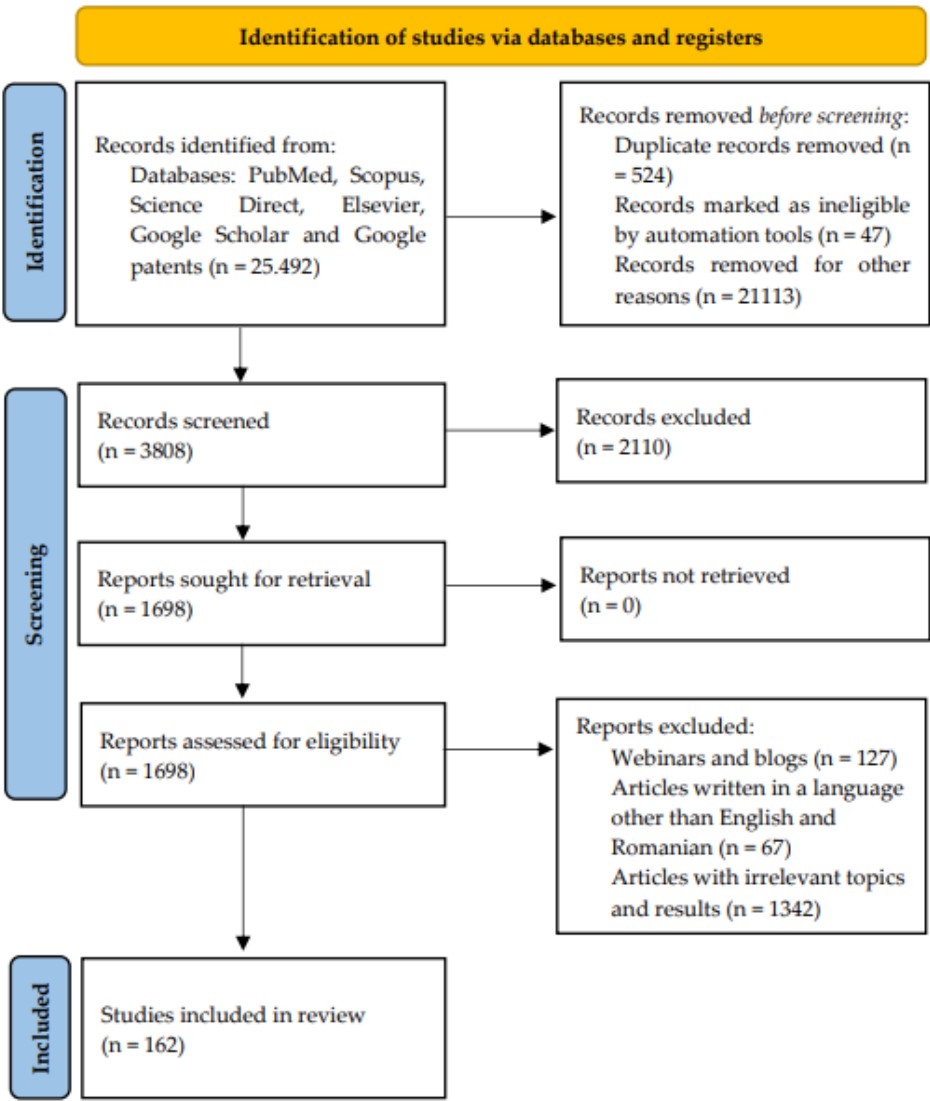

**Figure 1.** PRISMA 2020 flow diagram for the present review.

### 3. The Bioecology of the *P. spinosa* L. Shrub

*Prunus spinosa* L. is a perennial, thorny shrub, highly decorative for landscape and forest edges, belonging to the Rosaceae family, genus Prunus, representing one of the ancestors of *P. domestica* [6,23,26].

It is native to Europe (Figure 2), mainly central and southern Europe, except the lower half of the Iberian Peninsula, extending northwards to the southern part of the Scandinavian Peninsula [26,27].

*P. spinosa* L. is also widespread in western Asia and northwest Africa and is locally present in New Zealand, Tasmania, and eastern North America (USDA NRCS, The PLANTS database, 2015) [28], the Pacific Northwest and New England in the U.S. (Figure 2). Some authors believe that it originates in the northernmost tip of the European continent, in Scotland [29], and is commonly found in Europe, around deciduous forests, and in the temperate areas of Asia, especially in central, northern, western, and southern Anatolia. Towards the east, it reaches Asia Minor, the Caucasus, and the Caspian Sea [30]. Isolated populations have been found in Tunisia and Algeria. It is widespread in the Southern Alps, Switzerland, at altitudes up to 1600 m [31].

Species of *P. spinosa* L. are also found on the slopes of wild, uncultivated areas in several regions of Bosnia and Herzegovina [32]. It is commonly found at forest edges and openings, on sunny, rocky slopes, in ravines and river valleys, and meadows and pastures from low plains to mountains [33,34].

In Romania, *P. spinosa* L., can be found in lowland, plain areas but is more abundant in hilly areas, extending to mountainous areas with altitudes of 900–1000 m, being present at the edge of agricultural lands, decorating the landscape or on abandoned pastures, as well as on the edge of oak and beech forests [26,35–37].

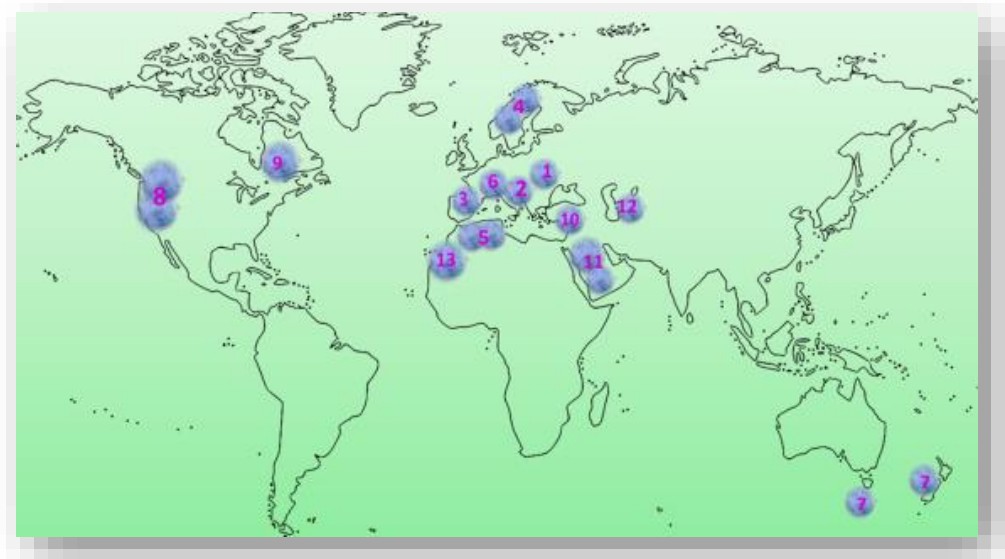

**Figure 2.** The habitat of *P. spinosa* L. (Plants of the World Online. Royal Botanic Gardens, Kew, 2023) [38]. 1—In Romania, it can be found in all areas; 2—It can also be encountered on the slopes of uncultivated areas in Bosnia and Herzegovina; 3, 4—Native to central and southern Europe, except the lower part of the Iberian Peninsula, it spreads towards the north, up to the south of the Scandinavian Peninsula; 5, 6—Isolated populations in Tunisia and Algeria, being also widespread in lower and higher areas, up to 1600 meters altitude in the Southern Alps of Switzerland; 7—Locally, it can be found in New Zealand, Tasmania; 8, 9—Locally, it can be found in the east of North America, the northwest of Pacific and New England in the United States; 10—Widespread in Western Asia, temperate regions of Asia—central, Northern Western and Southern Anatolia; 11, 12, 13—Widespread in Asia Minor, Caucasus, Caspian Sea and North and Western Africa.

It is a 2–3 m shrub with dark blue-violet bark and dense, stiff, spiny branches that grow well on clay, loam, sandy, calcareous, and well-drained soils and is recommended for its ability to improve degraded land. *P. spinosa* L. is a frost- and drought-tolerant species that develops well in sunny areas, where it benefits from exposure to light, as it is a thermophilic thorny shrub. It is also found on mesic to dry soils, on the edges of oak and beech forests, or the banks of rivers with willows and poplars, making it an unlimited source of berries—raw material for the food industry. It does not require any special care [4,26,39–41].

The leaves are oval, 2–4.5 cm long, and 1.2–2 cm wide with a serrated edge, and the flowers are white (Figure 3). They have five petals, are hermaphrodite, insect-pollinated, and possess vasoprotective, anti-inflammatory, diuretic, vermicide, detoxifying (blood purifying), and spasmolytic activities [4,39,40]. The first flowers appear in early to mid-March, depending also on the temperature, continuing until mid-April [35,36,42].

The fruits of the *P. spinosa* L. shrub are small, spherical, blackish drupes (Figure 3), about 10–12 mm in diameter, covered with blue bloom, and have therapeutic and functional properties [43]. The flesh is greenish yellow, adherent to the stone, with a pronounced astringent aroma due to the high tannin content and an acidic taste, which is why they can only be eaten fresh when overripe and in very small quantities [44]. Harvesting time is late autumn, in November, after the fall of the mist, due to the decrease in astringency. The fruits can be harvested even in winter because they persist well on the branches [2,27,38].

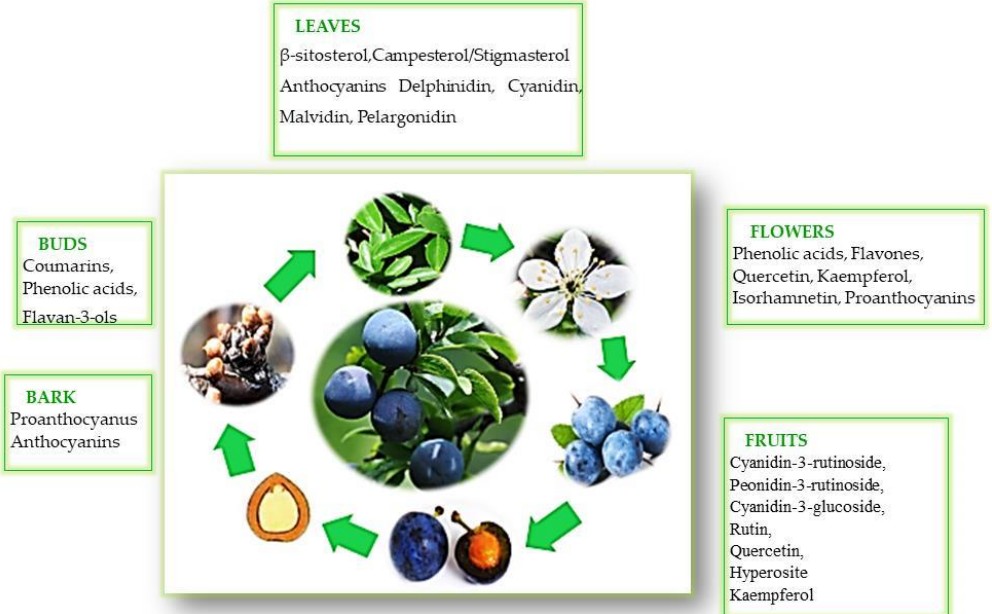

**Figure 3.** Different anatomical parts and phytochemicals content of *P. spinosa* L. Sources: [2,6,8,23,24, 45–47].

Data obtained to date have reported a total polyphenol and anthocyanin content contributing significantly to the anti-oxidant capacity of *P. spinosa* L. fruit based on the rich content of cyanidin-3-rutinoside (53.5%), peonidin-3-rutinoside (32.4%) and cyanidin-3-glucoside (11.4%) [48].

The fruits have proven functional effects on heart strengthening, in myocarditis and atherosclerosis [49,50]. Ethnopharmacological sources show that *P. spinosa* L. buds, popular in southern Europe, possess antihypertensive properties [51].

*Nutritional Values of the Blackthorne Fruits*

The nutritional composition and estimated energy value of the blackthorn fruits are presented in Table 1, which includes information acquired from various bibliographic sources.

**Table 1.** Nutritional value of blackthorn fruits based on literature.

| Reference | [52] | [37] | [15] | [53] | [54] |
|---|---|---|---|---|---|
| Energy (kcal/100 g) | 383.27 ± 7.09 | 57 | 154.93 | 249 | nd |
| Moisture (%) | 60.86 ± 1.69 | 54.85 ± 2.11 | nd | 69.37 | nd |
| Carbohydrates (g/100 g dw) | 88.51 ± 2.24 | 8.64 | 31.07 ± 0.62 | nd | 15.17 ± 25.83 |
| Proteins (g/100 g dw) | 2.86 ± 0.03 | 0.75 | 2.07 ± 0.04 | 3.4 | 0.99 ± 0.25 |
| Fat (g/100 g dw) | 1.98 ± 0.32 | 1 | 2.05 ± 0.12 | 2.06 | nd |
| Ash | 6.65 ± 2.03 | nd | 0.69 ± 0.04 | 2.72 | 1.18 ± 0.56 |
| Fiber (g/100 g) | nd | 9 | 5.79 ± 0.1 | 4.6 | 0.67 ± 0.26 |
| The place/period of blackthorn harvesting | The Natural Park of Montesinho territory, in Trás-os-Montes, North-eastern Portugal/September 2008 | Area of Cluj county, Romania/September 2019 | A mountain village Łącko., in the south of Poland/ns | Konya, Turkey/September 2003 | Kastamonu (Tosya) province, Turkey/ns |

nd—not determined; ns—not specified.

The observed variations in the nutrient content of *P. spinosa* L. fruits (as shown in Table 1) can primarily be attributed to climatic conditions. Fruits cultivated in countries with dry, hot climates are marked by a reduced level of moisture.

Based on the data included in Table 1, the quality of *P. spinosa* L. fruits has been evaluated with the aim of highlighting their nutritional-functional properties. At the same time, it has been emphasized that their integration into certain food products might result in metabolic benefits in the case of pathologies that are sensitive to the nutritional intake of biologically active macro and micronutrients.

For the judicious evaluation of the nutritional-functional quality of *P. spinosa* L. fruits, we performed the conversion of data on the content in macro and micronutrients into unitary measurement units (for macronutrients, g/100 g product, and micronutrients mg/100 g product). The identified macro and micronutrients [15,37,54–59] highlighted the fact that *P. spinosa* L. fruits may synergistically improve the functional capacity of the food products to which they are added.

The fatty acids found in *P. spinosa* L. fruits, including oleic acid, linoleic acid, arachidonic acid, linolenic acid, EPA (eicosapentaenoic acid), DPA (docosapentaenoic acid), and DHA (docosahexaenoic acid) [57] have the ability to improve the lipid profile. Thus, the ratio between hypocholesterolemic/hypercholesterolemic fatty acids (HH) [59] is favorable to hypocholesterolemic fatty acids. Therefore, this fruit might have a cardioprotective effect. Fatty acid content was dominated by monounsaturated fatty acids [57], followed by polyunsaturated fatty acids. According to the study conducted by Babalau-Fuss (2021) [57] on the analysis of fatty acid content in *P. spinosa* L. fruits (Table S1), it was found that monounsaturated fatty acids (MUFA) were the most abundant, accounting for 46.20% of the total fat content. Polyunsaturated fatty acids (PUFA) were identified in a proportion of 34.54%.

The report HH [57] is based on the fact that the fatty acids with pro-inflammatory effects, i.e., the myristic acid (C14:0) and the lauric acid (C12:0) were not identified in certain studies [15,37,54–59], while the palmitic acid (C16:0) is poorly represented [57]. This leads to an atherogenic index (AtIa) with anti-inflammatory potential, which is favorable to cardiometabolic pathologies [57].

The synergistic cardiometabolic effect could also be supported by the PUFAs/SFAs ratio, which was shown to have values of 1.83 [57] and 1.74 [52]. In another study, it had values of 0.23 [58]. One possible explanation for the discrepancy in this ratio is that *P. spinosa* L. is a perennial plant that develops in regions with temperate climates like Ukraine [45], Romania, and Portugal, rather than in regions with a continental climate like central Anatolia. Scientific literature indicates that temperature affects the various stages of growth and development in *P. spinosa* L. [60] species. Nevertheless, our current

understanding indicates a lack of extensive studies regarding the impact of climate types on the composition of fatty acids.

The higher this ratio, the greater the positive impact on cardiovascular human health of fatty acids intake [61].

Based on the consideration that nutrition is of great importance in the design, approval, and prescription of the nutritional quality of food, with proven effects on metabolic processes in certain food-caused pathologies, the HH, AtIa, and the health promotion index [61] was analyzed, in addition to PUFAs/SFAs indices [57,58], with the view of evaluating the nutritional value of dietary fats and the effect of fatty acids on cardiovascular pathologies (CVD).

The health promotion index [61] (HPI) is represented by the ratio between the sum of unsaturated fats and the sum of saturated fats (C12:0; C14:0, C16:0). Currently, it is mainly used in research relating to the nutritional quality of dairy products. The values of this index vary from 0.16 to 0.68 [61], compared to AtIa values, which are considered optimal at values ranging between 0.38 and 0.39. The identified value of the $\omega 6/\omega 3$ ratio (3.103) is in accordance with the cardioprotective dietary references [58]. From a psychosensory point of view, dairy products and pastry products rank among the foods in which *P. spinosa* L. fruits can be integrated with the best results. The inclusion of varying amounts of *P. spinosa* L. fruits, well tolerated by consumers, might significantly improve the nutritional-functional value of certain products since these fruits were associated with high cardiometabolic benefits [58,62].

The number of proteins identified [15,37,54–59] was comparable to that of other berries, presenting a potential to correct possible amino acid deficiencies, with a possible activity to reduce the limiting amino acid status in certain food formulae, especially lysine in cereals. The identified essential amino acids [63] were in amounts comparable to the reference values recommended by FAO by age groups.

An analysis of the amino acid composition of *P. spinosa* L. fruits [58] revealed that leucine, an essential amino acid responsible for regulating blood glucose and energy levels, was found in a concentration of 122.6 mg/100 g. This amount corresponds to 7.66% of the recommended daily allowances (RDAs) established by FAO (Food and Drug Administration) and WHO/DRIs (World Health Organization/Dietary Reference Intakes). These organizations recommend a daily leucine intake of 1600 mg for adults. Also, the quantity of Isoleucine detected at 99.2 mg/100 g in relation to the RDAs of 1400 mg/day can fulfill the required amount of this amino acid at a proportion of 7.085%. This amino acid plays a crucial role in the formation of hemoglobin and the regulation of energy. Additional essential amino acids detected in *P. spinosa* L. were Valine (87.8 mg/100 g), Phenylalanine (84.7 mg/100 g), Lysine (50.6 mg/100 g), and Threonine (47.6 mg/100 g). Tyrosine and aspartic acid are considered non-essential amino acids, meaning that their quantities are not relevant [58].

Differences were noted in the sugar content values (Table 1), ranging from 8.64 to 88.51 g/100 g. The variation in sugar content may be attributed to the intrinsic physicochemical properties and ripeness of blackthorn, as well as the environmental conditions [54].

Among the mineral elements identified [32,54,57], potassium had the highest amount (Table 2), followed by phosphorus, calcium, and sodium. The potassium levels varied between 1035.826 and 2014.23 mg/kg, the calcium levels ranged from 19.86 to 1504.41 mg/kg, and the sodium levels varied between 2.56 and 534.81 mg/kg (Table 2). Phosphorus was only determined by Babalau-Fuss et al., 2020 [57].

Such amounts ensure proportions of 93.55% K, 21.41% P, 12.15% Ca, and 26.74% Na, respectively, from the recommended daily amount [63]. Their nutritional importance is based on the metabolic effects since potassium represents the major cation of the intracellular space, while its extracellular amount has an important role in muscle activity, especially that of the heart muscle [63]. Together with Na and Mg, it plays a role in maintaining the electrokinetic difference at the level of cell membranes.

**Table 2.** Mineral content of *P. spinosa* L. from different data in the literature.

| References | [54] | [57] | [32] |
|---|---|---|---|
| Minerals elements | mg/kg | mg/kg | mg/kg |
| K | 2014.23 | 18,711.18 | 1035.82–1245.38 |
| Ca | 807.99 | 1504.41 | 19.86–34.23 |
| Mg | 188.1 | 972.21 | 8.57–11.83 |
| Na | 42.87 | 534.81 | 2.56–12.22 |
| Fe | 6.79 | 16.04 | 4.00–9.17 |
| Zn | 8.24 | nd | 0.35–1.80 |
| P | nd | 1511.37 | nd |
| Cu | 1.75 | nd | 0.93–2.45 |
| Mn | 1.44 | 4.58 | 0.80–2.38 |
| Al | nd | 26.33 | nd |

nd—not determined.

Calcium, due to its role as a cofactor in the blood coagulation process through the formation of thromboplastin and the transformation of fibrinogen into fibrin, is a key nutrient in a series of metabolic reactions of the human body [63]. It plays a role in maintaining normal muscle tone and the balance between muscle contraction and relaxation, and together with Mg, it reduces muscle excitability, thus justifying its use in the form of Ca and Mg salts in neuropsychiatry [63]. The amount of Ca and P identified [32,54,57] ensures a uniform Ca/P ratio, thus meeting the nutritional recommendations, which stipulate that this ratio should be 0.8–1. A proportion of 15% of the body's total phosphorus constitutes the metabolically active pool [63], present in the intra- and extracellular compartments, in the composition of nucleic acids, some coenzymes, and some lipids [63]. At the plasma level, it is present in free ionic form, bound to proteins, and in the form of sodium, calcium, and magnesium phosphates. Regardless of the existing quantity, all biogenic mineral substances are essential for the body to be able to synthesize or replace them. Vitamin C has been identified in amounts that provide 85.64% of the RDA, being also an absorption cofactor for non-heme iron present in plant-based products [57,63].

The utilization of *P. spinosa* L. fruits comes from their recognized efficacy in managing diarrhea, serving as both a diuretic and astringent [64]. These fruits can be preserved successfully by freezing or drying them, as they have a short harvest season and are sensitive to storage. *P. spinosa* L. fruits are frequently utilized in gastronomy, particularly in Turkish cuisine, for making jams or soaking them with sugar, honey, and bran to produce digestive and laxative liquor [64]. Drying these fruits allows their consumption throughout the year, making them suitable for use in different culinary preparations to enhance the nutritional and sensory qualities of certain foods. The dried fruit pulp of *P. spinosa* L. is regarded as a healthy snack due to its substantial concentration of phenolic compounds, antioxidants, and dietary fiber [64].

## 4. The Polyphenol Composition in Various Parts of the *P. spinosa* L. Shrub

Polyphenols are secondary metabolites with important and various functions in the plant world, representing an extensive group of phytochemicals. Table 3 includes data from the most recent literature (from the last few years) on the detection of phytochemical content identified in *P. spinosa* L. fruits, leaves, flowers, and branches.

**Table 3.** The phytochemical composition (phenolic acids, anthocyanins, and flavonoids) of blackthorn fruits, flowers, leaves, and branches is based on literature data from the previous few years.

| Organs of *P. spinosa* | Type of Sample/ Technique | Phenols | References |
|---|---|---|---|
| Fruits | Cold solution (1% BHT [*w/v*], 3% formic acid [*v/v*] in methanol) HPLC–DAD–MS | Phenolic acids<br>Cinnamic acid derivatives:<br><br>• 3-p-Coumaroylquinic acid;<br>• 4-p-Coumaroylquinic acid 1;<br>• Caffeic acid hexoside 1;<br>• Caffeic acid hexoside 3;<br>• p-Coumaric acid hexoside 1;<br>• 3-Caffeoylquinic acid;<br>• 4-Caffeoylquinic acid;<br>• 5-Caffeoylquinic acid 1;<br>• 3-Feruloylquinic acid;<br><br>Flavanols<br><br>• Catechin;<br>• Epicatechin;<br>• Procyanidin dimer 1;<br>• Procyanidin dimer 2;<br>• Procyanidin dimer 3;<br>• Procyanidin trimer 2;<br><br>Flavonols<br><br>• Quercetin triglycoside;<br>• Quercetin acetyl hexoside;<br>• Quercetin acetyl rutinoside;<br>• Quercetin hexosyl pentoside 2;<br>• Quercetin hexosyl rhamnoside;<br>• Quercetin-3-xyloside;<br>• Quercetin pentoside 2;<br>• Quercetin pentoside 3;<br>• Quercetin rhamnosyl hexoside;<br>• Querectin-3-galactoside;<br>• Quercetin-3-glucoside;<br>• Quercetin-3-rhamnoside;<br>• Quercetin-3-rutinoside;<br>• Isorhamnetin hexoside;<br>• Kaempferol pentoside hexoside;<br>• Kaempferol rhamnosyl hexoside 1;<br>• Kaempferol rhamnosyl hexoside 2;<br>• Kaempferol pentoside;<br><br>Flavones<br><br>• Apigenin pentoside;<br><br>Anthocyanins<br><br>• Cyanidin pentoside;<br>• Cyanidin 3-acetylglucoside;<br>• Cyanidin-3-glucoside;<br>• Cyanidin-3-rutinoside;<br>• Pelargonidin-3-glucoside;<br>• Peonidin-3-acetylglucoside;<br>• Peonidin-3-glucoside;<br>• Peonidin-3-rutinoside;<br>• Petunidin-3-rhamnoside. | [65] |

| Organs of *P. spinosa* | Type of Sample/ Technique | Phenols | References |
|---|---|---|---|
| Fruits | Ethyl acetate fraction of methanol-water extract (75:25, *v/v*) in dried fruit UHPLC-PDA-ESI-MS | Phenolic acids Protocatechuic acid 4-O-hexoside; Protocatechuic acida; 3-O-Caffeoylquinic acid; p-Hydroxybenzoic acida; Caffeoylshikimic acid derivative; Vanilloyl malate hexoside; 3-O-p-Coumaroylquinic acid; p-Coumaric acid O-hexoside; 5-O-Caffeoylquinic acid; cis-3-O-Feruloylquinic acid; 4-O-Caffeoylquinic acid; Caffeic acid 3/4-O-hexoside; 3-O-Feruloylquinic acid; Vanillina; 4-O-Caffeoylshikimic acid; 4-O-Feruloylquinic acid; Caffeoylshikimic acid; Caffeoylshikimic acid; p-Coumaroylshikimic acid; Aromadendrin hexoside; p-Coumaroylshikimic acid; Flavonols Quercetin 3-O-β-D-galactoside; Quercetin 3-O-(6″-O-α-L-rhamnopyranosyl)-β-D-glucopyranoside; Quercetin 3-O-β-D-glucopyranoside; Quercetin 3-O-α-D-xylopyranoside; Quercetin 3-O-α-L-arabinopyranoside; Quercetin 3-O-α-L-arabinofuranoside; Quercetin 3-O-(4″-O-β-D-glucopyranosyl)-α-L-rhamnopyranoside; Quercetin 3-O-α-L-rhamnopyranoside; Quercetin malyl-pentoside; Quercetin acetyl-hexoside-rhamoside. | [24] |
| Flowers | Defatted methanol-water extract RP-HPLC-PDA | Phenolic acids 3-O-Caffeoylquinic acid (neochlorogenic acid); 5-O-Caffeoylquinic acid (chlorogenic acid); 4-O-Caffeoylquinic acid (cryptochlorogenic acid); Caffeic acid; p-Coumaric acid; Flavanols (+)-Catechin; (−)-Epicatechin; Flavonols Kaempferol 3-O-α-L-arabinopyranoside-7-O-α-L-rhamnopyranoside; Kaempferol 3-O-β-D-xylopyranoside-7-O-α-L-rhamnopyranoside (lepidoside); Kaempferol 3,7-di-O-α-L-rhamnopyranoside (kaempferitrin); Kaempferol 3-O-α-L-arabinofuranoside-7-O-α-L-rhamnopyranoside; Kaempferol 3-O-β-D-xylopyranoside; Kaempferol 3-O-(4″-O-β-D-glucopyranosyl)-α-L-rhamnopyranoside (multiflorin B); | [66] |

**Table 3.** *Cont.*

| Organs of *P. spinosa* | Type of Sample/ Technique | Phenols | References |
|---|---|---|---|
| Flowers | Defatted methanol-water extract RP-HPLC-PDA | Kaempferol 3-O-α-L-arabinofuranoside (juglanin); Kaempferol 3-O-α-L-rhamnopyranoside (afzelin); Kaempferol 7-O-α-L-rhamnopyranoside; Kaempferol 3-O-(2″-O-E-p-coumaroyl)-α-L-arabinofuranoside-7-O-α-Lrhamnopyranoside; Kaempferol 3-O-(6″-O-α-L-rhamnopyranosyl)-β-D-glucopyranoside; Kaempferol 3-O-(2″-O-E-p-coumaroyl)-α-L-arabinofuranoside. Kaempferol; Quercetin 3-O-(6″-O-α-L-rhamnopyranosyl)-β-D-glucopyranoside (rutin); Quercetin 3-O-(2″-O-β-D-glucopyranosyl)-α-L-arabinofuranoside; Quercetin 3-O-β-D-glucopyranoside (isoquercitrin); Quercetin 3-O-β-D-galactopyranoside (hyperoside); Quercetin 3-O-α-D-xylopyranoside (reinutrin); Quercetin 3-O-α-L-arabinopyranoside (guaiaverin); Quercetin 3-O-(4″-O-β-D-glucopyranosyl)-α-L-rhamnopyranoside (multinoside A); Quercetin 3-O-α-L-arabinofuranoside (avicularin); Quercetin 3-O-α-L-rhamnopyranoside (quercitrin); Quercetin; | [66] |
| Leaves | 70% (*v/v*) aqueous-methanolic extract UHPLC-PDA-ESI–MS | Phenolic acids<br><br>• 3-O-caffeoylquinic acid (neochlorogenic acid);<br>• 3-O-p-coumaroylquinic acid;<br>• 3-O-feruloylquinic acid;<br>• 4-O-caffeoylquinic acid (cryptochlorogenic acid);<br><br>Flavanols<br><br>• procyanidin type-B dimer;<br>• procyanidin type-B dimer;<br>• (+)-catechina;<br><br>Flavonoids<br><br>• kaempferol 3-O-a-L-arabinopyranoside-7-O-a-L-rhamnopyranosidea;<br>• kaempferol 3-O-b-D-xylopyranoside-7-O-a-L-rhamnopyranoside (lepidoside);<br>• quercetin 3-O-(200-O-b-D-glucopyranoside)-a-L-arabinofuranosidea;<br>• kaempferol 3,7-di-O-a-L-rhamnopyranoside (kaempferitrin);<br>• kaempferol 3-O-a-L-arabinofuranoside-7-O-a-L-rhamnopyranosidea;<br>• quercetin 3-O-a-L-arabinofuranoside (avicularin);<br>• kaempferol hexoside-pentoside;<br>• kaempferol 3-O-a-L-arabinofuranoside (juglanin);<br>• kaempferol 3-O-a-L-rhamnopyranoside (afzelin);<br>• quercetin acetyl-hexoside-rhamnoside;<br>• kaempferol acetyl-hexoside-rhamnoside;<br>• kaempferol 7-O-a-L-rhamnopyranosidea;<br>• kaempferola;<br>• kaempferol 3-O-(2″-E-p-coumaroyl)-a-L-arabinofuranoside-7-O-a-L-rhamnopyranoside. | [67] |

**Table 3.** *Cont.*

| Organs of *P. spinosa* | Type of Sample/ Technique | Phenols | References |
|---|---|---|---|
| Branches | Lyophilized extract HPLC/MS | Phenolic acids<br><br>• Protocatechuic acid;<br>• Gallic acid;<br>• Caffeic acid;<br><br>Proanthocyanidins or flavan-3-ols<br>Ent-(epi)-catechin-(2α→O→7,4α→8)-(epi)-catechin-3′-O-gallate;<br>Ent-(epi)-afzelechin-(2α→O→7,4α→8)-(epi)-catechin-3′-O-gallate;<br>Ent-(epi)-gallocatechin (2α→O→7, 4α→8)(epi)-catechin;<br>Ent-(epi)-catechin (2α→O→7, 4 α→8)-catechin;<br>Ent-(epi)-gallocatechin (2α→O→7, 4α→8)-(epi)-catechin;<br>Ent-(epi)-catechin (2α→O→7, 4 α→8)-(epi)-catechin;<br>Ent-(epi)-afzalechin (2α→O→7, 4α→8) catechin;<br>Epigallocatechin;<br>Ent-(epi)-afzalechin (2α→O→7, 4α→8)-(epi)-catechin;<br>Gallocatechin;<br>Epicatechin;<br>Catechin;<br>Epiafzelechin;<br>Afzelechin;<br>Coumarins<br>5-hydroxy-6-methoxy-7-O-β-D-glucosyl coumarin;<br>5-hydroxy-6-methoxy-7-O-β-D-rhamnosyl coumarin;<br>Flavonols<br>Quercetin 3-O-rutinoside;<br>Kaempferol 3,7-O-dirhamnoside;<br>Kaempferol 3-O-arabinoside-7-O-rhamnoside;<br>kaempferol 3-O-arabinoside;<br>Quercetin;<br>Kaempferol | [6] |

BHT—2.6-di-tert-butyl-4-methylphenol; HPLC–DAD–MS—High-performance liquid chromatography with diode array coupled to mass spectrometry; UHPLC-PDA-ESI-MS—Ultra-high-performance liquid chromatography with diode array detection coupled to an electrospray tandem mass spectrometer; RP-HPLC-PDA—Reversed-phase high-performance liquid chromatography coupled with diode array detection; HPLC/MS—High-performance liquid chromatography coupled to mass spectrometry.

In the specialized literature, there are few recent comprehensive studies on the profile of the phenolic compounds identified in the *P. spinosa* L. plant [6,24,65–67]. From the studies presented in Table 4, it can be concluded that the solvents and the extraction method play an important role in highlighting the phytochemical composition of the plant. Thus, in the study by Magiera et al., 2022 [24], among all the compounds identified in *P. spinosa* L. fruits, neochlorogenic acid, and rutin were identified everywhere, regardless of the solvents used (Table 4). The highest amount of neochlorogenic acid (49.62 ± 1.51 mg/g dw) was reported in the extract in which solvents such as ethyl acetate fraction of defatted methanol-water were used, and the highest amount of rutin (2.93 ± 0.11 mg /g dw) was reported in the extract with N-butanol fraction of defatted methanol-water. At the same time, in the study by Petkovsek et al., 2016 [48], among the compounds identified in *P. spinosa* L. fruits (Table 4), anthocyanins represented the largest amount (2335.56 ± 185.79 mg/kg fw), followed by derivatives of cinnamic acids (487.79 ± 16.85 mg/kg fw) flavonols (278.57 ± 15.55 mg/kg fw), flavanols (225.97 ± 10.01 mg/kg fw) and flavones (4.15 ± 0.22 mg/kg fw). From the class of anthocyanins, Cyanidin-3-glucoside was predominant (1286.48 ± 116.07 mg/kg fw), from the class of cinnamic acids, 3-Caffeoylquinic acid (216.79 ± 7.87 mg/kg fw), from the class

of flavonols, quercetin pentoside 3 (70.59 ± 3.99 mg/kg fw), from the class of flavanols, procyanidin dimer 2 (100.64 ± 4.62 mg/kg fw), and from the class of flavones, the predominant was A pigenin pentoside (4.15 ± 0.22 mg/kg fw) [65]. The available data indicate that the anti-oxidant activity of *P. spinosa* L. fruits is primarily attributed to their high polyphenol and anthocyanin content. Specifically, the presence of cyanidin-3-rutinoside (53.5%), peonidin-3-rutinoside (32.4%), and cyanidin-3-glucoside (11.4%) has been found to contribute significantly to this activity.

In the case of *P. spinosa* L. flowers, the best solvent, with the most compounds identified, was represented by the combination of methanol and water (Table 4), and the predominant compound was kaempferol 3-O-α-L-rhamnopyranoside (afzelin), followed by quercetin 3-O-α-L-arabinofuranoside (avicularin), from the class of flavonols [66]. Among the compounds identified in the 70% (*v/v*) water-methanol extract of the dried *P. spinosa* L. leaf using UHPLC-PDA-ESI–MS, the predominant compounds were flavonoids, such as kaempferol 3,7-di-O-a-L-rhamnopyranoside (kaempferitrin) and kaempferol 3-O-a-L-arabinofuranoside-7-O-a-L-rhamnopyranosidea [67]. Also, the presence of isolated flavonoids with a high abundance of kaempferitrin and 3-O-a-L-arabinofuranoside-7-O-a-L-rhamnopyranoside, represents a unique characteristic of the *P. spinosa* L. plant [67,68]. Regarding the compounds present in the branches of *P. spinosa* L., Pinacho et al., 2015 [6] isolated phenolic compounds from air-dried branches (Table 4), then ground them into a fine powder and successively extracted by sequential cold maceration with dichloromethane, ethyl acetate, ethanol and water at room temperature in a closed container several times, evaporated and finally lyophilized. From the lyophilized extract obtained, 26 compounds were isolated, among which one was unidentified, and among those identified, phenolic acids, such as protocatechuic acid, gallic acid, and caffeic acid, were identified for the first time in the genus Prunus.

The focus of Volodymyr and Iryna's 2019 research was on examining the bioactive substances present in various *P. spinosa* L. plant anatomical parts [45]. Their findings revealed that the anthocyanin content in the bark ranged from 50 to 163 mg/100 g dry matter (dm). The leaves contained the highest concentration of catechins, ranging from 617 to 1031 mg per 100 g dm. The content of catechins in the bark was significantly lower (440–684 mg/100 g dm) compared to the content in the leaves.

The study conducted by Ciuperca et al. in 2019 [69] focused on extracting polyphenols from the branches of *P. spinosa* L. in Bacău County, Romania. The study found a polyphenol content of 2.97 ± 0.059 mg gallic acid equivalent/g of dry weight, as well as a tannin content of 0.90 ± 0.033 g tannic acid equivalent/g of dry weight.

Therefore, upon examining all the organs of the *P. spinosa* L. plant, it appears that the fruits have been the most thoroughly studied thus far. The fruits of *P. spinosa* L. possess the greatest number of identified phenol compounds in comparison to other organs of the plant [24,65,70].

## 5. The Effect of Bioactive Compounds Found in the Blackthorn Fruits in the Treatment of Various Diseases

Oxidative stress is one of the major pathological mechanisms that occurs during the inflammatory process, affecting the integrity of cells by destroying lipids, proteins, and nucleic acid [71]. It represents a disruption of the pro-oxidant/anti-oxidant balance with a key role in the development of many chronic diseases such as CVD, stroke, diabetes, some cancers, and certain neurodegenerative disorders. Anti-oxidant phenolic compounds possess at least one aromatic ring that includes hydroxyl groups, and the anti-oxidant capacity of these compounds is mainly due to their tendency to chelate metals [72].

Phenols and polyphenols are considered a group of compounds with the highest antioxidant activity due to their property to scavenge ROS, limit ROS production by inhibiting the activity of oxidative enzymes and the chelation of trace elements, and increase the efficacy of endogenous antioxidants [73]. The low redox potential of flavonoids, due to

proton donation, allows them to reduce highly oxidized free radicals such as peroxides, alkoxyl, hydroxyl, and peroxide radicals [74].

Polyphenols and anthocyanins have strong anti-oxidant activity, significantly reducing the harmful effects of free radicals that are produced by reactive oxygen species [75,76]. The total polyphenols and anthocyanins content in the powder extract obtained from blackthorns, dried at 30 °C, recorded values of 340.23 (mg GAE/g DW) and 180.2 (C3GE mg/g DW), respectively [77].

Different mechanisms have been proposed to explain the anti-oxidant activity of anthocyanins, such as the ability to scavenge free radicals [78], chelate metal ions, inhibit lipoprotein oxidation [79,80], and form complexes with DNA [81].

Data presented in specialized literature conclude that the fruits of the *P. spinosa* L. shrub manifest astringent, diuretic, antidiarrheal, antidysentery, and antibacterial effects and are recommended for stomach pain, diarrhea, dysentery, kidney disease, nephritis, biliary dyskinesia, common colds, whooping cough [82], respiratory disorders [49,83], cardiovascular pathologies [49], diabetes, obesity but also as digestive stimulants [83].

These nutritional and therapeutical properties have been attributed to the different bioactive molecular components of polyphenols and anthocyanins, identified in significant amounts in the blackthorn fruit [77]. The dominant classes of phenolic compounds present in the extracts of the blackthorn fruit [19] are hydroxycinnamic acid derivatives (44.4%), anthocyanins (32.7%), and flavonoid derivatives (21.1%), which is in agreement with data presented in recent literature [16,84,85].

One of the dominant classes of polyphenols in blackthorn fruit is anthocyanins [86]. The total phenolic content and unique polyphenols analyzed by HPLC–DAD of the *P. spinosa* L. fruit have shown high levels of rutin and 4-hydroxybenzoic acid, followed by gallic and transsynaptic acids [5]. Total polyphenol content was positively correlated with anthocyanin content and anti-oxidant activity of fruit juice [87], including significant phytosterol content with a significant impact on prostate cancer [2].

Table 4 presents the mechanisms of action of bioactive compounds identified in *P. spinosa* L. and their associated beneficial effects.

**Table 4.** Mechanisms of action and beneficial effects of nutritional-functional compounds of *P. spinosa* L.

| Bioactive Compounds | Health Effect | Main Outcomes | References |
|---|---|---|---|
| Flavonoids (Quercitin, Rutin) | Neurogenerative effect | -acetylcholinesterase inhibition<br>-monoamine oxidase inhibition<br>-↓ peroxyl radical capture and oxidation<br>-neurotrophic action<br>-maintenance of physiological functions of vital organs | [88,89] |
| Flavonoids | Cardiovascular effect | -inhibition of pro-inflammatory enzymes<br>-anti-atherosclerotic effects<br>-anti-atherothrombotic effects<br>-modulation of lipid metabolism,<br>-normalization of the LDL/HDL ratio,<br>-improving capillary permeability<br>-improved endothelial function, vasodilatory effects<br>-↑ release of nitric oxide and uncoupling of endothelial nitric oxide synthesis<br>-↓ oxidative DNA damage | [48,62,90–94] |
| Proantocianidine | | -modulates lipid metabolism,<br>-↑ anti-oxidant capacity of plasma,<br>- improve vascular functions<br>-↓ platelet activity | [48,95] |

**Table 4.** *Cont.*

| Bioactive Compounds | Health Effect | Main Outcomes | References |
|---|---|---|---|
| Cianidin-3-rutinoside | Cardiovascular effect | -improve lipid ↓ mechanisms, <br>-inhibition of lipolytic digestive enzymes <br>-inhibition of lipid absorption processes <br>-anti-oxidant activity on ROS, <br>-antiglycating activity <br>-cardioprotective activity mixed competitive inhibition of pancreatic lipase and pancreatic cholesterol esterase <br>-inhibition of cholesterol mycelial formation linked to primary and secondary bile acid <br>-inhibition of cholesterol mycelial absorption in the proximal jejunum. | [48,96,97] |
| Cianidin-3-glucoside | | -↑ tissue tolerance to ischemic injury <br>-↓ risk of cardiovascular disease, hypertension, <br>-capacity to scavenge ROS <br>-↓ oxidative stress, enhancing inflammatory responses | [96,98] |
| Cianidin-3-glucoside Cianidin 3-rutinoside | Diabetes and associated metabolic diseases | ↓ risk of diabetes and obesity <br>↓ postprandial glucose by inhibition of pancreatic $\alpha$-amylase and intestinal $\alpha$-glucosidase <br>-modulates postprandial blood glucose by inhibiting carbohydrate digestive enzymes <br>-↓ glucose transport in the small intestine. <br>-inhibit glucose uptake in colorectal adenocarcinoma epithelial cells | [98,99] |
| Phenolic acids | Anticancer effect | -cytotoxic activity on some cancer cell lines <br>-induction in vitro of endogenous anti-oxidant mechanisms <br>-modulation of Nrf2 transcription factors, a regulator of cellular resistance to oxidative damage, <br>-↓ disruption of the pro-oxidant/anti-oxidant balance with a key role in some cancers | [16,20,23, 100] |
| Anthocyanins | | -unchanged dietary absorption and incorporation into cells, <br>-major contribution to establishing anti-oxidant activity, reducing cancer risk | [2,15,67,90, 101,102] |
| Fitosteroli | | -anticancer activity on prostate cancer | [15] |
| Cianidin-3-glucozide | | ↓ cancer risk due to the ability to scavenge ROS | [100] |

↓—decrease; ↑—increase.

Several recent studies have shown that *P. spinosa* L. fruits, due to their high content of phenolic acids and flavonoids, including anthocyanins, flavonols, and flavones [5,16,23] have beneficial effects on the wound healing process [19,103], cytotoxic activity on some cancer cell lines [16,20,23] and selective inhibitory effect on the growth of some strains of potentially pathogenic bacteria [5,104]. Phenolic acids belong to a large group of polyphenolic compounds, which are the focus of current research due to their ability to prevent the development of cardiovascular diseases, cancers, and degenerative diseases whose incidence grows with age [105].

In studies conducted with the purpose of determining antioxidant and antimicrobial activity, it has been shown that the *P. spinosa* L. fruit extract (PSF) exhibits antimicrobial activity against potentially pathogenic Gram-negative and Gram-positive bacteria [106,107]. The study examining [108] the influence of *P. spinosa* L. and *P. padus* L. seed extracts with methanol and dichloromethane on 16 pathogenic bacteria led to the observation that the methanolic extract of *P. spinosa* L. showed good activity against all tested strains. The results of this study are also supported by the results of the comparative study of three

species, wild blackberry, cornus mas, and *P. spinosa* L. [80], where it was shown that *P. spinosa* L. has the highest antibacterial activity compared to the other two species.

The in vivo experimental investigation of oxidative stress induced by a high-fat diet in rats, supplemented with a minimal dose of streptozotocin, a naturally occurring nitrosourea derivative with chemotherapeutic effect, led to results demonstrating that the fruit extract of *P. spinosa* L. has a dose-dependent anti-oxidant capacity in both liver and brain [5]. It has also been shown to exhibit cellular anti-oxidant activity by inhibiting the hemolysis of human erythrocytes [5].

Based on the in vitro analysis of native primary polyphenols and phenolic metabolites found in the standardized flower extract from *P. spinosa* L., one study has emphasized their protective effects on human plasma components, especially on fibrinogen, isolated in human plasma matrix. It has also been shown that, based on the analytes tested in amounts of 1–5 g/mL in vivo, a significant reduction of structural changes of fibrinogen molecule under peroxynitrite-induced oxidative stress conditions has been obtained. In particular, a decrease in the oxidation and/or nitration of amino acid residues, including tyrosine and tryptophan, and the formation of high-molecular-weight aggregates [8] has been observed.

The leaf extracts showed antimicrobial, antidiabetic, and antitumor effects, representing an easily accessible natural source of bioactive compounds with potential application in food supplements and phytopharmaceuticals [109].

Experimental data presented in the specialized literature show that anthocyanins can exert therapeutic activities on diseases associated with oxidative stress, for example, coronary heart disease and cancer [110]. Specialized biochemical data have shown that anthocyanins are absorbed from the diet in an unchanged form [90] and are incorporated at the cellular level, both in the plasma membrane and in the cytosol [102], being one of the main classes of flavonoids that significantly contribute to the establishment of anti-oxidant activity with the potential to interact with biological systems, conferring antibacterial enzyme inhibition effects, cardiovascular protection, and overt anti-oxidant effects.

Blackthorn fruits are a rich source of antioxidants and can exert a strong protective effect on tartrazine-induced toxicity. The protective effects of *P. spinosa* L. fruit powder on hematological, biochemical parameters, and organic lesions in Wistar rats were analyzed after daily administration of the food additive tartrazine, dissolved in water, for 7 weeks [6,43]. Following this study, the results had the significant effect of improving the studied parameters.

Research that analyzed the ability of phenolic compounds to interact with biological systems that modulate gene expression was one of the postulated mechanisms to explain some of the health benefits of these compounds, which are found in *P. spinosa* L. fruits [111]. From a chemical point of view, phenolic compounds are molecules with one or more phenyl rings and one or more hydroxyl groups, capable of reducing reactive oxygen species conferring redox properties. The in vitro studies have demonstrated the strong anti-oxidant activities of most of these compounds [101], inducing endogenous anti-oxidant defense mechanisms by modulating transcription factors such as Nrf2, which is a central factor and regulator of cellular resistance to oxidative damage, thus being a therapeutic target in aging-related diseases [112,113]. These properties give phenolic compounds sanogenic properties, which are considerable in the prevention of chronic diseases associated with oxidative stress, such as cardiovascular and neurodegenerative diseases, diabetes, and various types of cancer [17,18].

*5.1. Neurodegenerative*

Studies evaluating the role of dietary antioxidants and the benefits of functional foods on neurodegenerative pathologies, such as dementia, Alzheimer's, and Parkinson's disease, have highlighted the fact that nutritive-functional foods can inhibit acetylcholinesterase, responsible for neurological disorders, and antioxidants maintain vital the physiological functions of organs, reducing oxidation by trapping the peroxyl radical [88].

The fruits of *P. spinosa* L. contain bioactive compounds in significant quantities. Through the specific anti-oxidant, anti-inflammatory, anti-apoptotic, and antithrombotic mechanisms of acetylcholinesterase and monoamine oxidase, they exert neuroprotective effects with the potential of neurotrophicity and maintenance of neurological health [89].

Recent studies have shown that polyphenols, including flavonoids, quercetin, and rutin, are well represented in *P. spinosa* L. fruits (Table 4), along with phenolic acids, and stilbenes, alkaloids, carotenoids, catechins, and terpenes have great potential in treating neurodegenerative disorders. Based on these studies, the multiple beneficial nutritive-functional effects of secondary metabolites on neurological health deserve special attention, as they demonstrate the ability to simultaneously act on different pathological targets and may help treat complex neurodegenerative disorders [89].

Antioxidants can modify oxidative stress and slow the symptoms of these neurodegenerative diseases [114]. Endogenous and dietary antioxidants can directly detect and neutralize ROS through an Nrf2-mediated mechanism and reduce the oxidation of damaged cellular molecules by inducing the synthesis of anti-oxidant enzymes to confer protection against oxidative stress [115]. A damaged anti-oxidant system can represent a favorable risk factor in the pathogenesis of diseases [116,117]. Reactive oxygen species (ROS) subject the brain to an acute oxidative stress insult. Thus, in Parkinson's disease, dopamine metabolism, mitochondrial dysfunction, and the neurotoxic effects of abnormal $\alpha$-synuclein accumulation promote ROS generation [118] in a context where a reduced activity of the endogenous anti-oxidant systems is present. In neurodegenerative disorders, anti-oxidant mechanisms are major ways in which phytochemicals exert their neuroprotective effects, such defense mechanisms being vital for neurological health.

Oxidative stress, an important mechanism involved in the pathogenesis and progression of many disorders, manifests similarly to pathogenic factors in a variety of neurological diseases [119]. Thus, it can be associated with their etiology. However, further research is needed in order to identify the mechanisms of selective neuronal vulnerability to ROS in different brain regions and to determine the effectiveness of anti-oxidant therapy on these diseases [120,121].

High oxygen consumption, low anti-oxidant levels, and low regenerative capacity have been identified as risk factors likely to induce oxidative damage in brain tissues [122]. Research on the therapeutic effect of dietary antioxidants on ROS has shown that they could alleviate neurodegenerative symptoms [114].

The main objective of contemporary scientific research is related to the identification of foods with increased content of substances with anti-oxidant potential on several diseases so that primary prophylaxis can be ensured through dietary intake, especially through the use of these functional foods over long periods, which are easily available, as part of any diet [123].

In this sense, combined therapies that relied on using polyphenols and memantine were analyzed due to the blocking effect of glutamate and green tea [124], which proved more effective in defending cells against cytotoxic injury and motor impairment than either ingredient used independently. Another combination therapy, [125] using catechins and acetylcholinesterase inhibitors, led to the conclusion that studies should primarily focus on combining medical and nutritional methods in order to develop the most effective therapeutic approach in the prevention of neurodegenerative diseases. However, the key strategy of these therapies must take into account the individual optimization of the intake of exogenous antioxidants for each patient, depending on the condition and function of the body.

*5.2. Dyslipidaemias and Associated Cardiovascular Diseases*

Antioxidants, due to their low volatility and high stability, can maintain the level of nutrients in food products [126]. However, for clinical assessment, it is necessary to measure the quantity of antioxidants in body tissues and fluids [23]. High levels of polyphenols and flavonoids have potential interventional value in several chronic conditions, such as

diabetes and cardiovascular diseases [127], but studies on human subjects have shown some disadvantages related to limited bioavailability, unknown optimal dose, or the appropriate timing of treatment. In this sense, studies on improving the bioavailability of antioxidants are ongoing [128].

An in vitro study on the bioactive potential of the *P. spinosa* L. flower extracts, which focused on the phytochemical profile of this plant and aimed to identify the mechanisms whereby these extracts influence cell safety, inhibit pro-inflammatory enzymes, and have protective effects against oxidative stress, showed that phenolic fractions have anti-oxidant capacity and might inhibit enzymes. Thus, it has been demonstrated that the extracts from blackthorn flowers show alternative potential as an ingredient for functional products that support the treatment of pathologies related to oxidative stress and inflammatory changes, especially in cardiovascular protection [90]. The flowers have a polyphenol content comparable to grape seed extracts, which attributes them functional properties on CVD.

Following the analysis of the total polyphenolic content [6] from the branches, leaves, flowers, and fruits of *P. spinosa* L., the blackthorn plant tissues were ordered as follows: branches > flowers > fruits ≥ leaves. The authors of this study claim that the flowers of *P. spinosa* L. could represent a promising functional ingredient for future research on the fortification of some food products with natural antioxidants.

In phytotherapy, blackthorn flowers are recognized as a food supplement due to their rich flavonoid content [129]. A total of 36 constituents, mainly flavonoids and phenolic acids, have been identified in the flowers of *P. spinosa* L., as opposed to only 25 observed in the leaves [67], 26 in the branches [6] and 29 in the fruits [85] of the same shrub. Due to such a complex phenolic matrix, blackthorn flower extracts have distinctive qualitative characteristics.

In the study on the impact of functional foods and nutraceuticals in the primary prevention of cardiovascular disease, Alissa and Ferns, 2012, analyzed the association between certain dietary patterns and cardiovascular health. Research on the cardioprotective potential of dietary components could support the development of functional foods and nutraceuticals in terms of cardiovascular protection so that consumption of flavonoid compounds together with the diet or through supplementation, may be associated with a reduced risk of cardiovascular events, as well as with reduced mortality [62].

In vivo studies have shown that flavonoids, as the most common group of polyphenols in the human diet, are richly represented in the flowers and fruits of *P. spinosa* L. and thus can have anti-atherosclerotic and anti-atherothrombotic effects in association with certain specific dietary patterns in cardiovascular pathologies, being safe in internal applications. In the studies in which the effects of nutrifunctional compounds administered individually, compared to dietary models of cardiovascular protection, were evaluated, it was observed that flavonoids in the early stages of the development of atherosclerosis due to the decrease in LDL oxidation, modulate lipid metabolism, normalize the LDL/HDL ratio, while also improving capillary permeability and endothelial function, with vasodilator effects due to increased nitric oxide release [62].

Both low molecular weight proanthocyanidins and phenolic acids have also been shown to modulate lipid metabolism, increase plasma anti-oxidant capacity, improve vascular functions, and reduce platelet activity in humans [93].

The anti-oxidant activity of polyphenols, due to their beneficial effects on CVD, has been evaluated in many studies [62,92,130], based on which it was decided to evaluate this activity in complementary chemical and biological models that reflect the most diverse and complex mechanisms. Following such evaluations, the result of the biological model confirmed the research hypothesis. Thus, the anti-oxidant activity of *P. spinosa* L. flower extracts may be crucial in understanding the beneficial effects on CVD in vivo.

Studies suggest that a sufficient intake of antioxidants may beneficially interfere with CVD by reducing ROS [91]. It has been shown [94] that antioxidants, administered either individually or in combinations, can modulate both the uncoupling of endothelial nitric

oxide synthesis and its function (eNOS) by scavenging free radicals or affecting specific pathways of radical generation, thus preventing oxidative stress and ameliorating vascular endothelial dysfunction (VED). Epidemiological evidence and dietary guidelines suggest that anti-oxidant-rich diets or anti-oxidant supplementation could maintain vascular health and prevent CVD.

Epidemiological studies documenting the beneficial effects of polyphenols and flavonoids on CVD [131] suggested that individuals with increased dietary intake of flavonoids and polyphenols have a much lower risk of cardiovascular disease. Anthocyanin supplementation reduced total cholesterol, triglycerides, and LDL-C (low-density lipoprotein) levels and increased HDL-C (high-density lipoprotein) levels in patients with dyslipidemia [132,133].

The data obtained so far [16,29,104,134] reported that *P. spinosa* L. fruits have a total content of polyphenols and anthocyanins that contribute significantly to anti-oxidant activity due to the rich content of cyanidin-3-rutinoside (53.5%), peonidin-3-rutinoside (32.4%) and cyanidin-3-glucoside (11.4%). Popović et al., 2020 [16] identified, apart from the anthocyanins mentioned previously, hydroxycinnamic acids (3-caffeoyl-quinic acid, 3-p-coumaroyl-quinic acid, 5-caffeoylquinic acid), flavonoids (quercetin-3-galactoside, quercetin-3-glucosides, quercetin-3-rutinoside, quercetin) and anthocyanin, peonidin-3-rutinoside.

Clinical evidence of pancreatic lipase inhibitory activity by cyanidin–3-rutinoside (C3R) is in agreement with previous reports suggesting that pure anthocyanins, cyanidin-3-glucoside (C3G), and peonidin-3-glucoside, and anthocyanins extracted from food sources demonstrated inhibitory action against pancreatic lipase, with the reduction of LDL cholesterol, triglycerides and the increase of HDL cholesterol [135].

The first report on the impact of (C3R on lipid-lowering mechanisms, including inhibition of lipid-digesting enzymes and absorption processes, showed that C3R acts as a binding-mode inhibitor so that it binds either to the active site of a free enzyme or to the enzyme-substrate complex [48].

Studies have shown that C3R is a natural anthocyanin, present in a significant proportion of the fruits of *P. spinosa* L., which possesses anti-oxidant, antiglycatory, and cardioprotective properties by inhibiting the digestion and absorption of lipids in vitro, being a mixed-type competitive inhibitor of pancreatic lipase, an inhibitor of pancreatic cholesterol esterase [95] and of primary and secondary bile acid-bound cholesterol mycelium formation [48] as well as their absorption in the jejunum proximal [96]. The inhibition of cholesterol esterase activity is due to the similar structure of the substrate of this enzyme to that of flavonoids, which allows them to bind irreversibly to the active site of the enzyme [136].

Cyanidin-3-glucosides, the third anthocyanin identified in *P. spinosa* L. extracts, have the potential to reduce the risk of dyslipidemia and associated cardiovascular diseases [135], obesity, diabetes, and cancer due to their ability to scavenge ROS and reduce oxidative stress, improving inflammatory responses [98].

In vitro studies that analyzed the bioactive potential of *P. spinosa* L. flower extracts [90] on cell safety and looked at their effect on the inhibition of pro-inflammatory enzymes have demonstrated the considerable anti-oxidant and enzyme inhibition capacity of phenolic fractions on pathologies related to oxidative stress that causes inflammatory changes. Therefore, these extracts can have beneficial effects on cardiovascular protection.

Special attention has also been paid to the anti-oxidant activity of the extracts from the branches, leaves, and fruits of blackthorn due to the possible mechanisms of action of the polyphenols present in the organs of this shrub [6,52,85]. The composition of blackthorn flower extracts seems promising in the context of CVD and demonstrates potential application in the food industry.

Polyphenols, including flavonoids, are specialized plant metabolites whose beneficial effects in CVD are commonly linked to their ability to influence two interdependent pathological processes of oxidative/nitrative stress and inflammation [137]. Polyphenols, being free radical scavengers, metal chelators, inhibitors of pro-inflammatory enzymes, and modifiers of cellular signaling pathways, can protect cellular and functional elements

of the circulatory system against lipid peroxidation, chronic inflammation, and oxidative DNA damage due to their vasodilator, vasoprotective, antiatherogenic, antithrombotic and anti-apoptotic effects [62,92]. Furthermore, reducing polyphenols and their metabolites can increase the total anti-oxidant capacity of blood plasma and tissue tolerance to ischemic injury [97].

*5.3. Diabetes and Associated Metabolic Pathologies*

C3R possesses anti-hyperglycemic properties, thus being able to modulate postprandial glycemia by inhibiting carbohydrate digestive enzymes and decreasing glucose transport in the small intestine [138].

The first report on the antidiabetic properties of *P. spinosa* L. leaf extracts was based on a study that tried to determine the polyphenolic profile and bioactivity of *P. spinosa* L. leaf extracts [109] with the aim of evaluating the total content of phenols, flavonoids, and anthocyanins with functional properties. The results of this study indicate that the examined samples had a notable potential in inhibiting $\alpha$-amylase and $\alpha$-glucosidase, enzymes related to type II diabetes.

The potential antidiabetic properties of *P. spinosa* L. fruits [104], evaluated on the basis of their ability to inhibit carbohydrate hydrolyzing enzymes, especially $\alpha$-glucosidase, were confirmed, being positively correlated with the results obtained by Popović et al., 2020 [16] who previously explored $\alpha$-amylase inhibitory activity ($\alpha$-AIA) and $\alpha$-glucosidase inhibitory activity ($\alpha$-GIA) of different *P. spinosa* L. genotypes. All examined samples had higher $\alpha$-GIA enzyme inhibition capacity than acarbose—a pseudo-oligosaccharide with a terminal C7-cyclitol that inhibits the action of enzymes that break down carbohydrates into simple sugars, slowing down the absorption of carbohydrates from the intestine and the glucose level becomes stable.

The study on the impact of *P. spinosa* L. flower extract on glycemic homeostasis [3], after 10 days of treatment with flower extract, led to the improvement of glucose homeostasis in hyperglycemic rats, an effect confirmed by the test for tolerance at the orally administered glucose. In addition, an increase of 49% in serum insulin concentration and an increase of 46% in serum $\alpha$-amylase activity has been observed after the period of 10-day intake of *P. spinosa* L. flower extract. Based on their study, the authors in [3] have claimed for the first time that *P. spinosa* L. flower extract has a potential in the management of hyperglycemia. This potential is based on the bioactive compounds present in *P. spinosa* L. flowers that can improve glucose tolerance and insulin secretion and decrease serum $\alpha$-amylase activity.

The results of the study on the impact of *P. spinosa* L. flower extract on blood glucose, glycemic load, and glycemic parameters on serum $\alpha$-amylase activity and serum insulin concentration [3] could serve as a guide for designing a nutraceutical mixture of polyphenols recommended in hyperglycemia support therapy.

*P. spinosa* L. is a rich source of phytochemical compounds and polyphenols, including phenolic acids and flavonoids, type A proanthocyanidins, anthocyanins, flavonols, and flavones, flavan-3-ols, showing anti-inflammatory, diuretic, purifying activities based on these compounds blood, spasmolytic and antitumor activities [77,90,139]. All these phenolic compounds have a strong anti-oxidant capacity [77] based on the results regarding the ability to regulate the hyperglycemic level, adipocytokine gene expression, and the improvement of some metabolic activities [99]. The hypoglycemic effects of *P. spinosa* L. on hyperglycemia include physiological mechanisms such as increased insulin sensitivity of peripheral tissue, inhibition of digestive enzymes involved in the breakdown of carbohydrates, and inhibition of glucose absorption in the gastrointestinal tract [140,141].

The inhibition of $\alpha$-amylase activities by the polyphenols present in *P. spinosa* L. has been demonstrated in several studies [141]. The increased insulin secretion in the hyperglycemic group [3] treated with the extract agrees with the in vivo results and confirms the action of the extract as a stimulator of insulin secretion.

The studies presented in the specialized literature [142] claim that quercetin, rutin, myricetin, and kaempferol, well represented in the fruits, flowers, and leaves of the *P. spinosa*

L. shrub, show properties that significantly improve insulin sensitivity and decrease glucose concentration in streptozotocin-induced diabetes. Among the flavonoids, kaempferol has the greatest influence on improving blood glucose levels, glucose tolerance, and insulin sensitivity [143].

The mixture of leaf and flower extract of *P. spinosa* L. has been shown to attenuate hyperglycemia and oxidative stress in streptozotocin-induced diabetic rats, as well as the severe degenerative and necrotic changes in diabetes [144], remarkably recovering β-cells. These results support the fact that there would be an alternative way to control high blood glucose levels, which is one of the most important complications of diabetes, protecting against oxidative stress and liver and pancreatic damage by increasing the anti-oxidant capacity in diabetes.

*5.4. Cancer Pathologies*

Karakas et al., 2019 [4] conducted tests to reveal the effect of the methanolic extract of *P. spinosa* L. fruits on cytotoxicity and cell viability, measured on different cancer cell lines in vitro, treated with different doses of 1–20 mg/mL methanol extract from *P. spinosa* L. fruit. Based on cell viability assays, significant cytotoxic effects of the methanol extract of *P. spinosa* L. fruit were observed in glioblastoma brain cancer cell lines but without any toxicity on pancreatic cancer cells. The numerous studies evaluating the anti-oxidant activity of polyphenols have concluded that these compounds can directly neutralize reactive oxygen species (ROS) and can act through a mechanism mediated by Nrf2 (Nuclear factor E2-related factor 2), leading in this way to the activation of the transcription of cytoprotective genes (phase II detoxification genes) and inducing the synthesis of anti-oxidant enzymes with protective properties against oxidative stress [47]. It is worth emphasizing here the process whereby the central regulatory factor of cellular resistance to oxidative damage (Nrf2) led to delayed replicative senescence of cells, being involved in the prevention and progression of cancers [115,145,146].

Analyzing the content of phytosterols and the antiproliferative activity of the ethanolic extracts of flowers, leaves, and fruits of the *P. spinosa* L. shrub on prostate cancer, it was shown that the highest amount of phytosterols was recorded in the ethanolic extract from the leaves, thus recommending further investigations on other cancer cell lines [2]. The in vitro evaluation of the antiproliferative activity of phytosterols showed that β-sitosterol, campesterol, and stigmasterol present in *P. spinosa* L. leaf extracts, even in low amounts, may be responsible for the inhibitory effect on the growth of cell lines involved in prostate cancer [2].

The quercetin present in the fruits, flowers, and buds of *P. spinosa* L. represents one of the beneficial flavonoids with significant antidiabetic, angiogenesis inhibition, cell cycle modulation, anti-oxidant, apoptosis-inducing, and anti-inflammatory bioactivities. Since most quercetin derivatives appeared to be involved in three important biological targets, namely the ARE signaling pathway, mitochondrial membrane potential stress, and the p53 stress response, it led to the examination of their internalization into the cell [47]. The obtained results showed that these compounds could enter the cell to carry out their biological activity, so the quercetin derivatives from the *P. spinosa* L. fruit extracts could accumulate in the intramitochondrial compartment, and the localization in the mitochondria is functional for the capture of ROS, mostly generated in this compartment, which allowed the measurement of the anti-oxidant capacity. The findings of the study conducted by Colomba et al., 2023 [47] confirmed a significant (maximum after 1 h), albeit transient (no visible effects at 18 h) anti-oxidant activity of the extract.

These nutrifunctional properties, analyzed in numerous in vitro studies, have shown that flavonoid compounds such as quercitin exhibit anticancer activities in a variety of cancer cells, such as MDA-MB-453 (breast cancer); U2.US/ MTX300 (osteosarcoma); HeLa (cervical cancer); U138MG (glioma); HT-29 (colorectal xenografts); CWR22Rv1 (prostate cancer). Quercetin has also been shown to block the growth and spread of melanoma and inhibit its metastatic potential [147]. Quercetin exerts anticancer activities by inhibiting

the G1/S or G2/M phases of the cell cycle, and the main cellular targets of quercetin are topoisomerase II, p27, p21, and cyclin B. Moreover, the free radical scavenging activity of quercetin can reduce cancer by recognizing three centers of reactivity [148].

Inhibition of tumor necrosis factor TNFα signaling by the anthocyanins delphinidin-3-O-glucoside and peonidin-3-O-glucoside, present in *P. spinosa* L. extracts [29,134] was observed in the study evaluating anti-inflammatory and signaling activity of TNF-α [98].

## 6. Applications of *P. spinosa* L. in Food Industry

The species of the *P. spinosa* L. shrub are used in the food industry and phytotherapeutic nutrition due to the significant content of biologically active substances present in the fruits, leaves, flowers, buds, and bark of this shrub, all of these having validated health benefits [2,8,149,150]. The high content of functional compounds in blackthorn makes them appropriate for use in the form of fruits, flowers, leaves, buds, or bark, as well as in the form of teas, infusions or decoctions, hemorrhages, kidney diseases, biliary dyskinesia, uremia, gout, diarrhea, growth, and development disorders as well as in body weight reduction therapies. Various preparations, such as juice and syrup, can be obtained from the blackthorn fruits based on the nutrifunctional properties correlated with the biochemical data [151], which allowed the highlighting of the significant content of antioxidants. These compounds, due to their high stability and low volatility, help to maintain the level of nutrients and preserve the texture, color, taste, aroma, freshness, and functionality of the products, so these are attractive to consumers. In addition, they can contribute to reducing the amounts of synthetic additive substances used in the food industry [14].

When performing the phytochemical analyses of the ethanolic extracts from the fruits and leaves of blackthorn, the presence of phenolic compounds was highlighted. The anti-oxidant and antimicrobial capacity of these extracts was also demonstrated, which recommends their use both in phytopharmacy and in the food industry [109,134,152]. Therefore, blackthorn can contribute to the sustainability of the food industry. At the same time, foods rich in polyphenols could have essential functions in preventing pathologies generated by oxidative stress and certain inflammatory processes due to their anti-inflammatory and antibacterial action [153].

Many studies state that *P. spinosa* L. fruits can be considered not only an "undiscovered" health-promoting type of food but also a potential source of pigments, flavors, and bioactive ingredients that can be used for the production of functional foods and nutraceuticals [43]. Due to the anti-oxidant and antimicrobial properties of polyphenols, blackthorn fruit extracts could be used as natural colorants and preservatives in the food industry [5,14,154]. The addition of blackthorn berries to ice cream improves quality parameters such as color and aspect, gum structure, and overall acceptability [11,155]. Contributing to the nutritional value and the functional and sensory properties of food, blackthorn fruit can be used as a functional additive in isotonic drinks, kombucha, or probiotic yogurt [10,156] as well as in fermented meat products [157]. *P. spinosa* L. fruits are a good source of fresh edible antioxidants and a sustainable raw material for obtaining syrups, juices, wine, liqueurs, and tinctures. Its dried fruits can be added to herbal teas along with other dried fruits.

The number of patents (Table 5) was developed based on blackthorn with various functional and therapeutic applications.

**Table 5.** List of patents based on the therapeutic and functional applications of *P. spinosa* L.

| Application No. | Species/Part | Results/Mechanism | References |
|---|---|---|---|
| WO 2016/157233 A1 | *P. spinosa* L.-fruits, *Trigno* variety | The composition is used, in particular, for the treatment of human tumors but also as a dietary supplement. | [158] |
| CN108576591A | *P. spinosa* L.-fruits | This invention is based on the use of *P. spinosa* L. as a raw material to make a dessert. Its long-term consumption contributes to the elimination of cough and has a protective effect on the lungs. | [159] |

**Table 5.** *Cont.*

| Application No. | Species/Part | Results/Mechanism | References |
|---|---|---|---|
| CN1970719A | *P. spinosa* L. fruits | The invention reveals a technique for preparing wine from *P. spinosa* L. fruits by the fermentation method. | [160] |
| FR2634783A1 | *P. spinosa* L. branches | This invention consists of obtaining a liqueur for the appetizer. The process consists of respecting the proportions of the various constituents, wine, alcohol, sugar, and branches of *P. spinosa* L., during controlled fresh maceration for 48 h. | [161] |
| EP3052202B1 | *P. spinosa* L.—flowers extract | The invention relates to obtaining a cosmetic product in which *P. spinosa* L. flower extract was used as a skin tanning agent. It helps to improve melanin formation in human skin cells by topical application. | [162] |

## 7. Conclusions

This study aimed to identify the bioactive, functional nutrients present in *P. spinosa* L., with a view of emphasizing how this health-beneficial plant can be used in various food formulae since nutrition plays a crucial role in the design, homologation, and prescription of the nutritional quality of specific food-types. Our analysis was also targeted at developing hypotheses and attributing meaning to the context, paving the way for the continuation rather than the cessation of research on the nutritional value of blackthorn, an aspect that can only take place through a development that cannot prevent the appearance of associative surprises.

Our review focused on data that stated the diversity of theoretical orientations or research practices and requested the possible synthesis in a methodological study appropriate to the level of theoretical development of the prospective approach regarding the beneficial biofunctional and metabolic effects of the studied *P. spinosa* L. fruits.

To introduce the fruits of *P. spinosa* L. into different food products, the data obtained in our review respected the criteria of objectivity and fidelity by which the results were compared with those obtained by different researchers or published by different authors, being repeatedly analyzed in detail.

Based on the data examined in this review regarding the beneficial properties of *P. spinosa* L., we consider that both the fruits and the other parts (flowers, leaves, buds, and bark) present a real potential for use in the food industry for the processing of natural functional foods, or for obtaining new nutraceutical products. At the same time, they can be used as additives, natural colorants, or preservatives in food processing.

**Supplementary Materials:** The following supporting information can be downloaded at: https://www.mdpi.com/article/10.3390/horticulturae10010029/s1, Table S1: The fatty acids from *P. spinosa* L. fruits.

**Author Contributions:** Conceptualization, S.I.V. and M.F.B.; methodology, A.R.M.; formal analysis, C.A.R.; investigation, A.I.A., D.D. and R.B.; writing—original draft preparation, M.F.B. and C.A.R.; writing—review and editing, A.R.M. and M.F.B.; visualization, R.P.; supervision, S.I.V.; project administration, M.F.B.; funding acquisition, S.I.V. All authors have read and agreed to the published version of the manuscript.

**Funding:** This research was funded by the University of Oradea, within the Grants Competition "Scientific Research of Excellence Related to Priority Areas with Capitalization through Technology Transfer: INO-TRANSFER–UO", Project No. 309/21.12.2021.

**Data Availability Statement:** Not applicable.

**Conflicts of Interest:** The authors declare no conflict of interest.

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
