# Peer review of "An Overview of the Phytochemical Composition of Different Organs of Prunus spinosa L., Their Health Benefits and Application in Food Industry"

_horticulturae, doi:10.3390/horticulturae10010029_

Round 1

Reviewer 1 Report

Comments and Suggestions for Authors

This review explores the potential of Prunus spinosa L. for incorporation into the food industry for the development of novel food formulations. The review focuses on examining its chemical and phenol composition, along with its potential positive effects on human health. A thorough analysis of 152 articles, selected through an inclusion process, has been conducted. Various parameters, including bioecology, nutritional profile, mineral content, and phenol composition have been evaluated. Finally, a well-conducted review of its effects on human health was conducted. Based on the findings, I recommend the publication of this work after taking into consideration the following minor corrections:

-        Figure 1: Include the name of the databases used for the articles’ search.

-        Please give the references used for the statements in: lines 92-94, 111-114, 126-133, 143-146, 200-202.

-        Figure 3: Cite the references used for obtaining the information present in this graph.

-        Line 156: Do not use an independent numeral subsection if it is the only one.

-        Line 174: In the sentence “The identified fatty acids…” describe what compounds are being referred to.

-  Line 178: Could be given information about the content of these monounsaturated/polyunsaturated fatty acids?

-        Line 185-187: What is the reason for this difference in the PUFAs/SFAs ratio?

-        Line 174-205: A very extensive narrative on fatty acids content is given. It would be recommendable to include a table compiling the main results obtained in different articles regarding the presence and the content of these compounds in the P. spinosa L. fruit, to a better understanding.

-        Line 206-210: It is said that “…presenting a potential to correct possible amino acid deficiencies, with a possible activity to reduce the limiting amino acid status in certain food formulae…” but the amino acids profile is not given. Please describe the presence and content of amino acids in P. spinosa L. fruit. Also give the content of essential amino acids.

-        Line 2011-212: It is said that the highest value for sugars was 15.171g/100g but in table 1 is specified 88.51 [37] as carbohydrates, please specify.

-        Line 215-216: Check the values for the highest content minerals, also the complete sentence’ sense.

-        Line 243: Also, the branches are studied. Add.

-        Table 4: Check the spelling (second group of bioactive compounds). Make clearer the sub-divisions in the table to distinguish between the bioactive compounds and their healthy effects.

Comments on the Quality of English Language

Minor editing of English language required: check the spelling and few sentences' meaning (described in the review report)

Reviewer 2 Report

Comments and Suggestions for Authors

Ref 2763504

The authors state that the objective of this review is to analyze prospective approaches that emphasize the beneficial biofunctional and metabolic effects of different botanical parts of Prunus spinosa L. for human health. The authors aim to conceptualize innovative approaches by which Prunus spinosa L. fruits can be incorporated into certain food products to take advantage of their potential metabolic effects on cardiovascular and other pathologies related to antioxidant intake.

The work presented by the authors is clear, properly organized, well written and uses the PRISMA methodology for the search, selection and analysis of the articles included in the review. Before it can be accepted for publication some issues must be addressed.

Specific comments:

As a result of the emergence of chronic non-communicable diseases, there are two directions: 1) the return to natural and traditional products, minimally processed, and 2) the production of functional foods, often using materials or unconventional additives. Please add a little more context to increase the strength of your review.

Lines 40-48. Authors should be careful about how readers may interpret statements. While it is true that evidence shows that the consumption of foods containing phenolic compounds (plant kingdom: fruits and vegetables) modulates and prevents degenerative diseases, the statements should not be specifically focused on antioxidant capacity per se. It is known that the consumption of antioxidants is also related to pro-oxidant effects in the body. The combination of vitamins, minerals, water, fiber and phenolic compounds, among others, have shown biological effects against various pathologies in humans. It is precisely the change in global dietary patterns (decrease in the consumption of fruits and vegetables and increase in the consumption of ultra-processed, high-energy-dense foods, added with sugar and salt) that currently explains most non-communicable chronic diseases. Please include some context so that your manuscript has greater strength and scientific relevance.

Line 75 (Research Methodology). Please clarify in the manuscript the date range used in the information search. Please describe the criteria for inclusion and exclusion of the articles analyzed. Indicate the sections (subtitles) that were made in this review.

Figure 3. Please indicate where the information related to the content of phenolic compounds in the botanical parts was obtained.

Table 2,3… Please homologize the units used. Choose the most appropriate one (mg/kg or microgram/gram).

In section 3 some text should be added regarding gastronomy and the consumption of P. spinosa L. as food and as a medicinal plant, including its botanical parts, so that section 4, the objective of the review and the conclusions make more sense.

Please homologize the use of the terms “fruit,” “Blackthorne” and “blackthorn fruits” in detailing nutritional composition, or indicate the botanical parts to which it refers (e.g. Line 157 and 160).

The text mentions both “P. spinosa L. fruits” and “P. spinosa L. Fruits”. Please check the use of capital letters.

Line 211. Check the amount “15,171 g/100 g”. The authors use the “,” and the decimal point “.” interchangeably. Please homologize to avoid confusion. If choosing to use “,” as the decimal separator, do not place more than two decimal places. Correct this in the manuscript for tables and text.

Table 4. Please review the word “Falvonoids”.

Reviewer 3 Report

Comments and Suggestions for Authors

This paper reviews the identification of bioactive and functional nutrients of Prunus spinosa L. that can be incorporated into diverse food formulations and their impacts on metabolic processes in specific diet related pathologies. The topic is interesting and is worth to be published after minor revision.

1.     The data about moisture content in Table 1 is unbelievable.  

2.     The measurement units and the effective decimal places of data should not be differed in one table such as Table 2.

3.     The structure of Table 3 is not so reasonable because there are a lot redundant  or repeated descriptions.
